# The Tripods in Daoist Alchemy: Uncovering a Material Source of Immortality

**Zhen Fan**

Faculty of Education, University of Macau, Macau SAR 999078, China; yb97111@um.edu.mo

**Abstract:** The tripod (*ding* 鼎) and the nine tripods (*jiuding* 九鼎) are significant in ancient China, appearing often in Daoist alchemy. However, they have been largely ignored by the scholarship on Daoism. Early Daoist alchemy saw the tripod and the nine tripods as critical elements in the production of immortality, but the outer alchemy (*waidan* 外丹) gave up refining the outer elixir by tripod due to technical reasons. The tripod was merely mentioned in the elaboration of outer alchemy. Later, in the Southern Song dynasty, inner alchemy (*neidan* 內丹) rebuilt the significance of the tripod and the nine tripods in inner refining, inventing new theories, such as the body-tripod metaphor, the nine orbits, and the lunar phases. This paper outlines the history of the (nine) tripods as a concept and implement in Daoist alchemy.

**Keywords:** Daoist alchemy; tripod (*ding* 鼎); nine tripods (*jiuding* 九鼎); inner alchemy; Bai Yuchan 白玉蟾; *Jinye huandan yinzheng tu* 金液還丹印證圖





## 1. Introduction

The tripod and nine tripods are significant in ancient China. In 606 BC, Wangsun Man 王孫滿, a royal descendant of the Zhou dynasty (1046–256 BC), outlined a tale, which recounted that the tripod originated from the nine tribes and was offered to the Xia dynasty as the heaven's will ([Yang 1995](), pp. 669–72). In 113 BC, the tale was expanded when a tripod was uncovered in Fenyin 汾陰 ([Tang 2015](); [Chao 1975](); [Li 2001](); [Yang 2019](), pp. 278–79). An official of Emperor Wudi integrated the tale of Yu 禹, Fuxi 伏羲, and Huangdi 黃帝 into the invention of the tripod.[1] They are the founders of three periods, namely, Three Emperors (*sanhuang* 三皇), Five Monarchs (*wudi* 五帝), and dynastic history. Later, a group of *fangshi* 方士 (alchemists) claimed that Huangdi became immortal after he cast tripods ([Sima 2014](), p. 1674).[2] Thereafter, the tripod gradually changed from a symbol of state power and heaven's will to a sign of immortality.

Daoists inherited much of the *fangshi*'s thoughts connecting between Huangdi and the tripods. For instance, Ge Hong 葛洪 (283–343) received three secret alchemical scriptures: *Taiqingdan jing* 太清丹經 (Scripture of the Elixirs of Great Clarity), *Jiudingdan jing* 九鼎丹經 (Scripture of the Elixirs of the Nine Tripods), and *Jinyedan jing* 金液丹經 (Scripture of the Elixir of the Golden Liquor) ([Ge 1996](), p. 71). Ge Hong named the second secret alchemical scripture as *Huangdi jiudingdan jing* 黃帝九鼎丹經in the hagiography of Zhang Ling ([Ge 2010](), p. 190).[3] In these scriptures, there is a noticeable change: the number of tripods cast by Huangdi has increased from three to nine.

These changes in symbolic and numerical understanding have become the primary knowledge background for Daoism to refine elixirs by tripod. State power, the will of heaven, and immortality symbolized by the tripod were absorbed by Daoist alchemy from the beginning, becoming important research topics in Daoist alchemy.

Modern scholarship on Daoist alchemy began from studies of the history of science.[4] The earliest paper that discussed the tripod was written by the chemist Cao Yuanyu in 1933 ([Cao 1933](); [Barnes 1934](), [1936]()). But both Cao and Joseph Needham's team tried to match the ancient objects and skills with modern chemistry based on irrelevant literature ([Ho and

Needham 1959; Ts'ao et al. 1959; Needham et al. 1980, pp. 1–167), which led to errors. Chen Guofu made a further contribution to the literature by clearly indicating the time when the important works on alchemy were written (Chen 1954, 1979, 1983, 1997). More specifically, he noted the time when tripods of different shapes appeared in medieval Daoism. Later, Zhao Kuanghua characterized the essential implements and work principles of alchemy (Zhao and Zhou 1998, pp. 390–415).

Taken together, Cao Yuanyu, Needham, Ho Ping-yu, Chen Guofu, and Zhao Kuanghua analyzed the principles of elixir refinement through the use of tripods with distinct approaches. They studied the materials and structure of the alchemist tripod by analyzing diagrams from ancient texts and testing the results through experiments. However, these findings have limitations. For instance, since the chemists paid much attention to the refining process of the elixir and its raw materials, they ignored the cultural symbolism and the rituals related to the tripods. Two scholars, Fabrizio Pregadio and Han Jishao, noted this issue in their books (Pregadio 2006; Han 2009), but they only gave brief descriptions on the cultural symbolism of the tripods. Eugene Wang analyzed the cyclical time of the First Emperor's Tomb by a tripod and other sculptures, exhibiting a cosmology of inner alchemy that may have existed in the Qin dynasty (221–207 BC) (Wang 2014).[5]

In previous studies, historians of science have pointed out that the outer-inner tripod (*neigui* 內櫃 and *waigui* 外櫃) and water-fire tripod (*shuihuo ding* 水火鼎) were the main facilities used for refining the elixir. Chen Guofu and Zhao Kuanghua agreed with this statement, confirming that the outer–inner tripod appeared in the Tang dynasty (618–907) and the water–fire tripod in the Song dynasty (960–1279) (Zhao and Zhou 1998, p. 400). These tripods were different from the traditional tripods of the bronze age, which had ears and legs. Instead, they were designed primarily for refining elixir. Nevertheless, also during the Song dynasty, these functional tripods were quickly abandoned by a novel school of inner alchemy (*neidan* 內丹).

While Daoist researchers regard inner alchemy as a distinct religious practice, almost all of them admit that the cosmology, theory, and early terminologies of inner alchemy were mainly developed from outer alchemy (Zhang 2001, pp. 23–32; Pregadio and Skar 2000, pp. 464–72). However, when inner alchemy thrived in the eleventh century, it chose the traditional tripod of the Pre-Qin period with double ears and three legs. In fact, the refining facilities of outer alchemy included the crucible (*fu* 釜), altar (*tan* 壇), furnace (*lu* 爐), and cabinet (*gui* 匱). All of them are very different with the traditional tripod. Inner alchemy invented a series of new symbols for the tripod by returning to the traditional shape.

This article tries to outline the role of the tripod from outer alchemy to inner alchemy.

## 2. The Relationship among the Tripod, Elixir, and Crucible in the Outer Alchemy of Medieval Daoism

The first issue to be clarified in medieval alchemy is the relationship between the tripod and the elixir. Ge Hong received the *Jiudingdan jing* in the fourth century, but the scripture hardly introduces the meaning of the tripod in the title. The word most frequently used in this scripture is *jiudan* 九丹, which refers to the elixirs that are refined through nine different cyclical transformations and have nine different effects. *Jiudan* is the most important elixir prescription in the second alchemical scripture Ge Hong received. He also mentioned the "Great Clarity Elixir" (*taiqing shendan* 太清神丹) in another prescription, in which he designated the "Ceremony of the Nine Tripods" (*jiuding ji* 九鼎祭) as a ceremony of the Elixir of Nine Tripods.[6] This proves that the "nine tripods" could in fact be the Nine Elixirs (*jiudan* 九丹). More evidence can be found in another alchemical scripture of *Huangdi jiudingdan jing*, which states that "the nine tripods are the nine elixirs 九鼎者，九丹也" (*Jiuzhuan Liuzhu Shenxian Jiudan Jing* 1988, p. 434b). To sum up, the full name of the nine elixirs should be the "Elixir of Nine Tripods" (*jiuding dan* 九鼎丹).

Moreover, before the end of the Eastern Han dynasty, there was no book saying that Huangdi had nine tripods. In Ge Hong's records, the number of Huangdi's tripods increased from three to nine. Moreover, the first founder of tripod, Yu, is no longer

associated with tripod in outer alchemy. There are only two scriptures which include Yu in the titles across the *Daoist Canon* and neither of the two scripts mentions the tripod or the nine tripods.[7] The reasons why Yu lost his status in Daoist alchemy are twofold. First, both Yu and the nine tripods have been closely associated with Confucianism since the Eastern Zhou dynasty. Second, Huangdi obtained his divine power from Daoism. As such, Daoism minimized Yu's presence and transferred the relation between Yu and tripod to Huangdi. In the sixth chapter of *Baopuzi*, *Weizhi* 微旨, Ge Hong changed Huangdi's tale of "melted and cast into a cauldron at the foot of Mt. Jing" (鑄鼎于荊山下) recorded in *Shiji* into "fly the nine elixirs at the foot of Mountain Jing and Tripod Lake于荊山之下，鼎湖之上，飛九丹成" (Ge 1996, p. 129). Later, in the Tang dynasty, this tale was changed into "fly the nine elixirs of nine tripods 飛九鼎九丹" (*Huangdi Jiuding Shendan Jingjue Jiaoshi* 2015, p. 53). In another scripture entitled *Zhouyi cantong qi* 周易參同契, the first elixir was also "named as the first tripod 名曰第一鼎兮" (*Zhouyi Cantong Qi* 2014, p. 356). Therefore, Pregadio translates the *Jiudingdan jing* as the "scripture of the nine elixirs" (Pregadio 2006, p. 197).

The tripod did not exist in the practice of the outer alchemy. In *Taiqing danjing*, the tripod could be used as a container for melting the great clarity elixir (Ge 1996, p. 77). According to *Wuling danjing* 五靈丹經, Ge Hong introduced the manufacturing method of the Elixir of Instant Success (*lichen dan* 立成丹). In this method, the alchemist first had to obtain copper by heating realgar and then cast a ware. This description does not indicate the type or shape of this ware, but it was by no means a tripod (Ge 1996, p. 79). In 2018, archaeologists excavated a vessel of so-called alum water (*fanshi shui* 礬石水) in a Western Han tomb of Luoyang, which was initially thought to have been refined through the method documented by Ge Hong. It is a great discovery in the study of outer alchemy, but the container is a bronze jar, not a tripod (Jiang et al. 2019). Another case comes from the hoarded treasures of the Tang dynasty excavated in Hejia Village, Xi'an. There are three kinds of vessels used for heating elixir. Two of them look like pots, while the third one has a handle, an explicit change to the shape of tripod (Figure 1). According to the alchemical materials found inside and the inscription written on the vessels, "*nuanyao*" (暖藥, warming elixirs), these vessels were used for refining elixir (Qi et al. 2003, pp. 159–71).

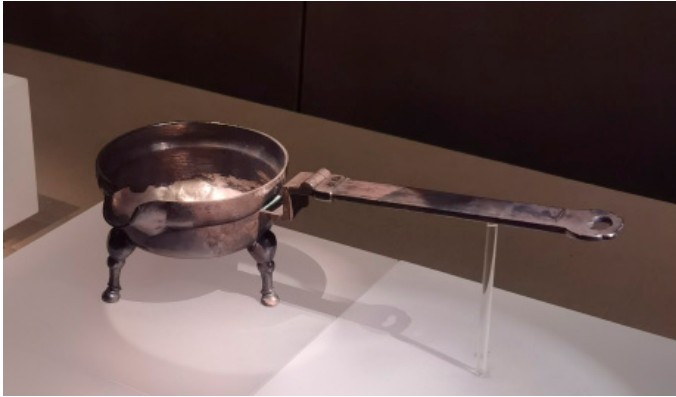

**Figure 1.** A silver pan (*cheng* 鐺) of the Hejiacun Tang treasures. Photographed by the author.

Despite the frequent usage of "nine tripods" in many elixirs, outer alchemists largely ignored the tripod.[8] Furthermore, these alchemists sometimes use *fu* (釜, caldron or crucible) instead of the "tripod" as the name of the boiling vessels. This *fu* is not made from bronze. Instead, *fu* is made from pottery, stoneware, or gypsum, according to the descriptions in *Huangdi jiuding shendan jing*, which include the "red mud crucible" (*chitu fu* 赤土釜), the "black, yellow, and red mud crucible" (*xuanhuang chitu fu* 玄黃赤土釜), and the "six and one mud crucible" (*liuyi nifu* 六一泥釜) (*Huangdi Jiuding Shendan Jingjue Jiaoshi* 2015, p. 10, 12, 14; Pregadio 2006, pp. 169–70). As Pregadio indicated, alchemical *fu* is mainly made from "red clay" (*chishi zhi* 赤石脂) (Pregadio 2006, p. 101). Another scripture entitled *Taiqing jing tianshi koujue* 太清經天師口訣 (DZ883) written before 650 AD (Pregadio 2006, pp. 54–55), a

little bit later than *Baopuzi*, provides more details on how to make clay crucible, identifying the raw materials to be used, such as red clay, vinegar, and the white bark of "Quercus dentata" (*hushu baipi* 斛樹白皮). The craft in making the clay crucible include steaming, cooking, kneading, smashing, drying, filing, griddling, and lacquering (Zhang 1988, p. 3). However, different from pottery and porcelain, heating is not necessary in the making of the clay crucible to prevent leakage. It is not until the Tang dynasty that porcelain alchemical crucibles were manufactured in kilns in Dingzhou, Chuzhou, and Hunan (Chen 1997, pp. 34–37).

The shape of the crucible or tripod in outer alchemy took on different forms. In *Huangdi jiuding shendan jingjue*, an isolated of iron "tripod" (鐵銚釪) was manufactured to support the crucible,[9] which replaced the original structure of the tripod. Unfortunately, there is no archaeological evidence of the crucible or iron tripod.[10] The followers of Ge Hong drew a few diagrams in Song dynasty (960–1276, Figure 2); however, none of the diagrams drawn in Song dynasty or modern times bear any resemblance to the traditional bronze tripods.[11] Furthermore, there is no evidence that alchemists used nine tripods in alchemical activities or related rituals. In fact, almost all the historians of science who care about Daoist alchemy have detected an aqueous solution (*shuifa* 水法)—which dominated the whole craft after the Tang dynasty (Ts'ao et al. 1959). *Shuifa* has one specific space for storing water on the bottom and another space for storing raw materials in the upper part. In contrast, the traditional tripod only has one space.

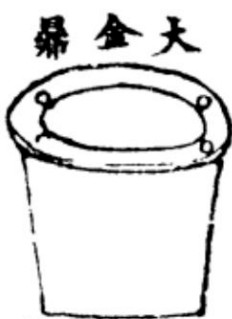

**Figure 2.** The golden tripod of the heavens (Bai 1988b, p. 165b).

In medieval Daoism, outer alchemy changed the understanding of the tripod that was prevalent during the Han dynasty. The meaning and function of the tripod were replaced by the elixir and crucible. Outer alchemy invented new refiners with higher efficiency by relinquishing the shape, structure, and symbolization of the tripod. Eventually both the tripod and the nine tripods came to carry no symbolic meaning, being simply enumerated in formulas along with other ingredients and tools.

### 3. The Body-Tripod Metaphor of Inner Alchemy

Even though the outer alchemists abandoned the tripod during the medieval age, it continued to be a valuable resource for Daoism. When inner alchemy originated at the middle Tang Era,[12] the concept of tripod was utilized anew.

Scholars agree that inner alchemy inherited the cosmological significance of tripod from Zhouyi cantong qi (Pregadio and Skar 2000, pp. 466–68). Zhouyi cantong qi is an early alchemical scripture that is even older than the three alchemical scriptures which Ge Hong received. Scholars discern that Zhouyi cantong qi has a different ideology compared to Ge Hong and it served as the most important scripture of inner alchemy (Skar 2003, pp. 164–71). It contains a special chapter named "Song of Tripod" (*Dingqi ge* 鼎器歌), which was written earlier than the other parts of the scripture. The Song of Tripod does not describe as many technical details as Ge Hong did, but it depicts a cosmological system built around the tripod. Therefore, Zhouyi cantong qi and the Song of Tripod became

the best location for inner alchemists to find ancient resources to facilitate and bolster the theory for their cultivations.

The concept of the tripod was also appropriated by inner alchemy. Zhang Guangbao argued that the biggest difference between inner alchemy and other inner practices of Daoism is that inner alchemy incorporates many aspects of outer alchemy into its theory (Zhang 2001, pp. 61–63). Thus, as one of the most visible elements of outer alchemy, the tripod entered the symbolic system of early inner alchemy quite quickly. For example, Luo Gongyuan 羅公遠, a pioneer of inner alchemy who lived in the Tang Xuanzong period (712–756), portrayed the tripod using the human body as a metaphor:

> My body is like the tripod. I press my right foot with my left foot and my body with my two hands, leaving the abdomen empty. Such that my body resembles the three legs of the tripod. 吾之身象鼎焉。以左足壓其右足，以左右手按其身，復虛，如鼎三足焉. (Zeng 2016, p. 302)

As I discussed previously, medieval Daoism gave up the traditional tripod with three legs and two ears. However, mainstream outer alchemy did not deter Luo Gongyuan from conceiving of the tripod in the image of the human body, especially comparing hands and legs to tripod's three legs. In that sense, Luo drew from terms in outer alchemy to enrich the early theory of inner alchemy by the simple use of analogy.

As inner alchemy developed more fully, most of its cultivators disregarded the simple analogy of Luo Gongyuan because they were pursuing eternity and found the tangible body trivial. At the end of the Tang dynasty, Zhongli Quan 鍾離權 (lived in the Late Tang dynasty, see Li 2004), the first master of inner alchemy, built up its cosmology into a complete system. However, he and his famous disciple, Lü Dongbin 呂洞賓 (may lived in the Five Dynasties and Ten Kingdoms to the Early Song dynasty, see (Ma 1986)), did not address the body–tripod metaphor. *Zhong-Lü chuandao ji* 鍾呂傳道集 (DZ263.14)[13] reports that Lü Dongbin once asked how to imagine himself when refining the elixir in his mind. Zhongli Quan depicted a shape of tripod in the following terms:

> In this case, visualize the central vessel as a tripod or a cauldron, either yellow or black, shaped like a carriage wheel 其想也，一器如鼎如釜，或黃或黑，形如車輪. (Kohn 2020, p. 179; Zhongli and Lü 2015, p. 113)

Here, the whole body is not metaphorically treated as a tripod; instead, a tripod "resides" in the cultivator's body or mind. Discussing the tripod and cauldron (as well as the crucible) together also shows that the early inner alchemy borrowed terms from outer alchemy. But the wheel shape of tripod was a temporary invention.

Inner alchemists after Zhongli Quan, such as Chen Tuan 陳摶 (ca. 871–989) and Zhang Boduan 張伯端 (ca. 987–1082), made huge progress in some specific fields of inner alchemy, such as cosmology, the source of elixir, and the theory of life and nature (*xingming* 性命). However, the body–tripod metaphor was at a standstill, though they used this metaphor frequently. A version of *Wuzhen pian* 悟真篇 (DZ236.26) includes a set of diagrams at the beginning of the scripture. One of the diagrams is named "Suspending Fetus Tripod" (*xuantai ding* 懸胎鼎, Figure 3) and depicts a tripod with three legs, just like the traditional tripod (Ye 1988, p. 713a). The description included in the diagram states:

> The Suspending Fetus Tripod has a circumference of about 50 cm (1 *chi* 5 *cun*); its width is about 16.665 cm (5 *cun*) in its middle part; and its height is about 40 cm (1 *chi* 2 *cun*) in total. It takes the shape of *penghu* (a kind of pot) and also looks like the human body. The tripod is separated into three parts as a metaphor for the Three Powers (*sancai*). It has a straight belly, the top, middle, and bottom are equal. The tripod goes 26.664 cm (8 *cun*) deep into the furnace but does not touch the ground. Therefore, it is named as Suspending Fetus, as well as the Cinnabar Tripod. Zhang Sui has noted that it has another name; that is, the Sacred Censer of the Great One (*Taiyi shenlu*). 鼎周圍一尺五寸，中虛五寸，長一尺二寸。狀似蓬壺，亦如人之身形。分三層應三才。鼎身腹通直，令上中下等均勻，入鑪八寸，懸於竈中不著地，懸胎是也。又謂之朱砂鼎。張隨注雲：又名太一神爐。

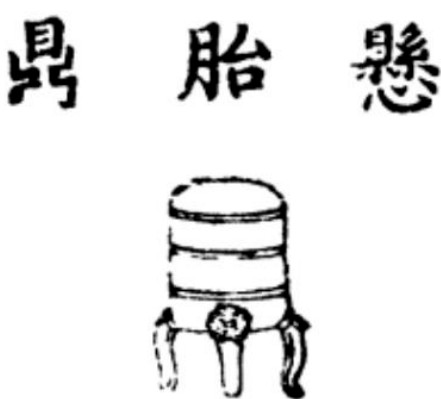

**Figure 3.** The Suspending Fetus Tripod in *Danfang baojian zhi tu* ([Zhang 1988](), p. 713a).

The description complies with many technical rules of outer alchemy. It is hard to judge whether the diagram or the description came first, but the latter offers many metaphors to imagine and explain the shape of tripod. The first is *penghu* 蓬壺. As a common concept in Daoism, *penghu* is used to represent Penglai Immortal Island (蓬萊仙島), which is shaped like a pot.[14] *Peng* is also a concept that compares *fang* 方 and *yin* 瀛.[15] Perhaps it indicates the lotus pod, but the explanation I offered above also stated that the Suspending Fetus Tripod is similar to the human body. Then, the writer provides the second metaphor: the three parts of the tripod signify the three main elements of the universe (*sancai* 三才): heaven, earth, and humanity. Furthermore, the so-called Suspending Fetus means that it has the legs to enter the oven and furnace, which prevents the main part of tripod from touching the flames directly. Taking into consideration this technical description, Cao Yuanyu and Needham cited the diagram when they discussed the implements of outer alchemy ([Cao 1933](); [Needham et al. 1980](), p. 17), but this diagram belongs to *Danfang baojian zhi tu* 丹房寶鑑之圖, which was used in annotating the inner alchemical scripture *Wuzhen pian*. Before the diagram of Suspending Fetus Tripod, the writer also drew the diagrams of the *Yin* and *Yang* (陰陽), dragon and tiger, true land (*zhentu* 真土), mercury, and lead to explain the theory of inner alchemy.[16]

Nevertheless, other versions of *Wuzhen pian* do not have the *Danfang baojian zhi tu*, which belongs to a book series called *Xiuzhen shishu* 修真十書 (ten books for cultivating the immortal). Scholars believe that the collection was edited in the Yuan dynasty (1271–1368) and the diagrams were drawn by Ye Wenshu 葉文叔 (died in 1161), who is the first annotator of *Wuzhen pian* too ([Fukui 1999](), pp. 245–71; [Hussein 2005b](), pp. 816–17). His diagram of the Suspending Fetus Tripod mixed the imagination of inner alchemy and technical concerns from outer alchemy, suggesting the rise of a new synthesis in the understanding of the tripod in the inner alchemy of Song Daoism.

## 4. The Nine Tripods in Bai Yuchan's Outer and Inner Alchemy

This synthesis was also expounded by Bai Yuchan 白玉蟾 (born in 1134, see [Gai 2013](), pp. 417–67), the alchemical master of the Southern Song dynasty, who is the founder of the Southern Lineage of the Golden Elixir (*jindan pai nanzong* 金丹派南宗). Generally, this lineage is regarded as an inner alchemical school, but it is easy to ignore that it advocated the cultivation of inner and outer alchemy at the same time ([Gai 2013](), pp. 817–49).

Bai Yuchan's only extant scripture of outer alchemy is *Jinhua chongbi danjing mizhi* 金華沖碧丹經秘旨 (DZ914).[17] It introduces the raw materials, required equipment, necessary prescriptions, and the intricate process of refining elixir. In addition, there are series of diagrams that illustrate what kind of tripod should be employed in refining the "Reverted Elixir in Nine Cycles" (*jiuzhuan huandan* 九轉還丹). Before the diagrams, Bai Yuchan presents a manual detailing how to make the "outer tripod" (*waiding* 外鼎), in which he uses the term "outer" (*wai* 外) to distinguish from the inner chamber of the refiner ([Bai 2013a](), p. 107). The "outer tripod" is made of porcelain. It can be fitted into an inner

chamber named *shenshi* 神室 along with other auxiliary materials, and altogether hung on the stove to refine elixir. According to the manual, the tripod is more complex than the traditional tripod, bearing a closer resemblance to the Suspending Fetus Tripod of *Danfang baojian zhitu*. Looking at the eleven diagrams of *Jinhua chongbi danjing mizhi*, none of them could be classified as the traditional tripod. However, there may be two exceptions. The Fifth Reverted Elixir, *Sanqing zhibao dan* 三清至寶丹 employs an inner chamber with three legs (Figure 4). Here, the author characterizes the inner chamber as the "inside tripod" (*neishi ding* 內室鼎) (Bai 1988b, p. 166a), but he also calls the entire diagram "Picture of Tripods" (*dingqi tu* 鼎器圖). The same thing happens in the *Wuyue tongxuan dan* 五嶽通玄丹, the Seventh Reverted Elixir (Bai 1988b, p. 166c–167a). Its diagram calls a porcelain tripod that exhibits an extra pedestal with three legs (Figure 5). These diagrams of *Sanqing zhibao dan* and *Wuyue tongxuan dan* conform with the manual to make the "outer tripod." Moreover, in another diagram, Bai Yuchan described the pedestal with three legs as *zeng* 甑 (Figure 6). He indicated that the device is used for placing the tripod (Bai 1988b, p. 162a).[18] In other words, this three-legged pedestal is not tripod.

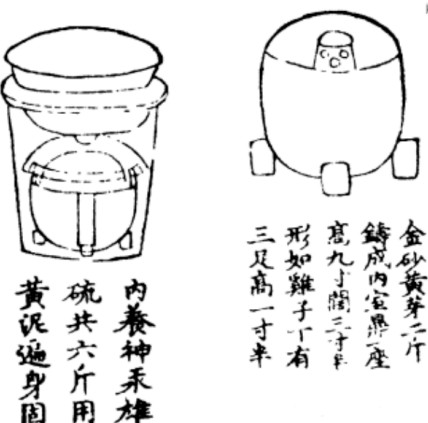

**Figure 4.** The Fifth Reverted Elixir of the Three Clarities' Treasure (*Huandan diwu zhuan sanqing zhibao dan* 還丹第五轉三清至寶丹) and the diagram of its refiner (Bai 1988b, p. 166a).

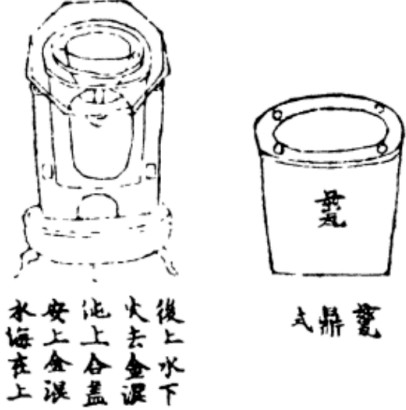

**Figure 5.** The Seventh Reverted Elixir of Pervading Mystery in the Five Great Mountains (*Huandan diqi zhuan wuyue tongxuan dan* 還丹第七轉五嶽通玄丹) and the diagram of its refiner (Bai 1988b, p. 167a).

*Jinhua chongbi danjing mizhi* shows that Bai Yuchan inherited the achievements of outer alchemy and "decorated" (修飾) them with his theories of inner alchemy (Meng 1993, p. 286). In other scriptures, Bai's definitions on the tripod are all from the angle of inner alchemy, such as:

Water source then soil crucible, crucible then golden tripod, golden tripod then jade furnace, jade furnace then sacred chamber, sacred chamber then core altar 水源即土釜，土釜即金鼎，金鼎即玉炉，玉炉即神室，神室即元坛. (Bai 2013a, p. 168)

To refine the form, the cultivator has to use the body as the state, use the heart as the emperor, and use the essence as the people. The form is the furnace, and the head of the cultivator is the tripod. Because the head is full of the essence, it must be refined by fire 鍊形，以身為國，以心為君，以精為民。形者，鑪也。首者鼎也。精滿於腦，故用火鍊. (*Zazhu zhixuan pian 1988*, p. 614; *Bai Xiansheng Qiongguan Zazhu Zhixuan Ji 1271*, p. 3.4a)

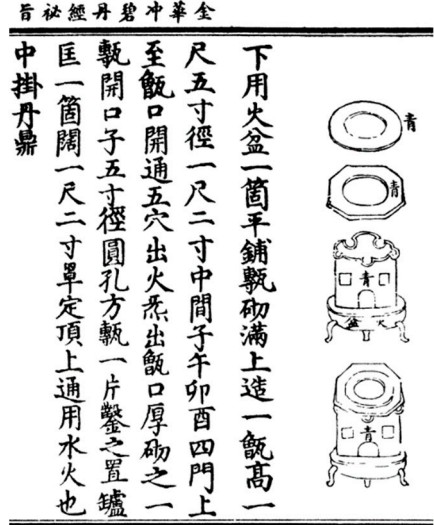

**Figure 6.** Picture of *Zeng* 甑 (Bai 1988b, p. 162a).

The first reference comes from *Gousuo lianhuan jing* 钩锁连环经 , which was gathered in *Haiqiong chuandao ji* 海瓊傳道集 (DZ1309) by Hong Zhichang 洪知常, in the Southern Song dynasty. This scripture quotes Taiyi Yuanjun 太乙元君 to illustrate the procedures of cultivating the elixir in body and mind. The word "*ji* 即" denotes the next step in the cultivation. The whole scripture is a rhyme that interprets the alchemical loop from the essence of nature, such as mercury and lead, to the elixir of none and chaos. Within this loop, the crucible, tripod, furnace, chamber, and altar are equal in Bai's eyes. That is, they are not trivial facilities in his inner alchemical theory.

The second reference establishes a concrete comparison to the human body. The reference was drawn from an article penned by Chen Nan 陳楠 (died in 1213), *Yinfu sui* 陰符髓,[19] which is a commentary on the *Huangdi yinfu jing* 黃帝陰符經 (DZ31). *Huangdi yinfu jing* is a brief scripture that sets forth the nature of the universe (Rand 1979; Ma 2012), and it could be decoded and interpreted in many ways. *Yinfu sui* unscrambles the *Huangdi yinfu jing* through the theory of inner alchemy, arranging the refiners of outer alchemy to fit specific human organs, rather than a general metaphor of the body. The Imperial Cabinet Library of Japan (*Naikaku bunko* 內閣文庫) has a version of the *Yinfu sui* with a preface showing that it was written earlier than 1244 (*Bai Xiansheng Qiongguan Zazhu Zhixuan Ji 1271*, p. 1.4a). Furthermore, *Yinfu sui* displays a close relation with *Yinfu jing sanhuang yujue* 陰符經三皇玉訣 (DZ119), which was cited in 1191 (Ren 1991, p. 91; Schipper and Verellen 2004, p. 696). On the other hand, *Yinfu sui* is included in *Zazhu zhixuan ji* 雜著指玄集 (DZ263a) as an article that is one of the *Xiuzhen shishu* 修真十書, a book series edited by a follower of the Southern Lineage of the Golden Elixir at the end of Southern Song dynasty (Pang 2019, pp. 46–54). Therefore, the notion that there is a one-to-one correspondence between different human organs and alchemical facilities was produced before 1161 and was accepted by the Southern Lineage of the Golden Elixir.

In addition to *Yinfu sui*, there is a suite of diagrams titled *Jindan huohou tu* 金丹火候圖 painted by Bai Yuchan in the first volume of *Zazhu zhixuan ji*.[20] One of the diagrams named *Jinding tu* 金鼎圖 (picture of the golden tripod) represents the tripod as a human body. It has two notes in both sides of the diagram (Figure 7) that read:

> The mind with nine orifices is called the golden tripod. That is the nine tripods made be the Yellow Emperor. 心有九竅謂之金鼎，黃帝鑄九鼎者，此也. (*Bai Xiansheng Qiongguan Zazhu Zhixuan Ji* 1271, p. 1.2a)[21]

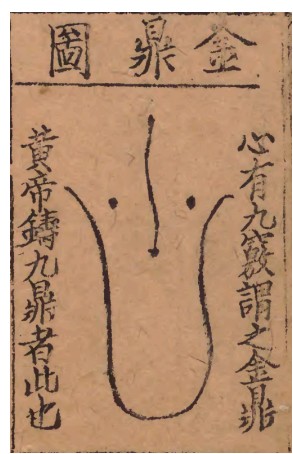

**Figure 7.** Picture of the Golden Tripod.

The diagram depicts two nipples[22] and a line that accesses the center of human body. The notes provide an explicit analogy likening the nine significant apertures of the human heart to golden tripods, as well as to the nine tripods of Huangdi. This is an innovation to the understanding of the nine tripods and Huangdi post Ge Hong.

In fact, Bai Yuchan repeatedly mentioned the nine tripods:

> The Yellow Emperor cast the nine tripods to refine golden elixir, and Laozi laid the foundation of the three mountains to build the sacred chamber. 黃帝鑄九鼎 以制金丹，老君基三山以創神室. (Bai 2013a, p. 239)

> Generally speaking, Daoists should not only do routines like burning incense in the morning, lighting lamps at night, ringing the bell, or living on a vegetarian diet, but they should also . . . After ten months, the fetus becomes full and the fire of nine tripods is sufficient. The cultivator will travel into the tunnel and view the landscape, fly and float in the sky. He will learn with heaven and tour with the Creator. 蓋知乎道士者，非止於晨香暮燈，板粥鐘齋而已，要當 . . . . . . 十月胎 圓，九鼎火足，乘隧扇景, 策空駕浮，與天為徒, 與造物者遊. (Bai 2013a, p. 240)

> Study and be amazed by the nine tripods in the three mountains. Learn to fast and restrain starvation. 學三峰九鼎奇，習休糧與閉饑. (Bai 2013b, p. 323)

The Yulong Wanshou Daoist Monastery (玉隆萬壽宮), located in Nanchang, Jiangxi province, is known as a place that is dedicated to the worship of Xu Xun 許遜, an immortal who appeared in the Northern and Southern dynasties (420–589). Toward the end of the Southern Song dynasty, the monastery belonged to the Jingming School, a novel Daoist school (Kan'ei 1978, pp. 80–89; Xu 2014, pp. 341–60). Bai Yuchan mentioned the nine tripods two times when he visited the Yulong Wanshou Daoist Monastery according to *Yulong Wanshou Gong daoyuan ji,* probably written in 1218.[23] At the beginning of the article, Bai Yuchan regards Huangdi casting nine tripods and refining golden elixir as the origins of Daoism, paralleling with the deeds of Laozi 老子. Bai's second reference to the nine tripods can be seen in a reply to Luo Ruoxu 羅若虛, the abbot of Yulong Wanshou Daoist Monastery at that time.[24] Luo delineated the history of Daoists serving the state and he thought that the most prosperous time of Daoism was the period of Zhenghe to Xuanhe (1111–1125),

because every Daoist acquired an official identity. However, Bai Yuchan disagreed with this argument and contended that Daoists should not merely perform regular worship, but instead they should cultivate themselves first. If they can achieve this, instructing followers is acceptable. In his view, the full fire in the nine tripods is what certifies the cultivation of a Daoist. Thus, Bai Yuchan deemed inner alchemy to be indispensable for the cultivation for a Daoist follower and the nine tripods as vital in the certification of a Daoist.

The manual of cultivating the nine tripods is actually one of the three or four treasures in the Southern Lineage of the Golden Elixir. Bai Yuchan's two main disciples recorded the inheritance of the manual among Chen Nan, Bai Yuchan, and their disciples:

> (The master) altogether imparted to us the Book of Nine Tripods, Golden Lead, and Sand Mercury, the Mnemonic of the Gram, Fire Sign, and Golden Liquid of the Great One, and the Article of the Purple Sky, and Roaring Lightening and Storm. 荷相授以九鼎金铅砂汞之书、太乙刀圭火符金液之诀、紫霄啸命风霆之文. (Bai 2013a, p. 364)

> The Master (Bai Yuchan) studied Daoism under the Master Chen Niwan (also known as Cuixu), received the Secret of the Gram of the Great One, *Book of the Nine Tripods and the Golden Elixir*, Method of Long Living and Sight, Article of Roaring Lightening and Storm in the Purple Sky, and the Means of Invisibility and Flight for Entering the Nothing. 先生师翠虚陈泥丸先生而学道焉，得太乙刀圭之妙，九鼎金丹之书，长生久视之术，紫霄啸命风霆之文，出有入无飞升隐显之法. (Bai 2013a, p. 376)

> Peng Si wrote a poem to express his profound impression of the nine tripods as well:

> Covered in golden sand, the nine tripods look bright. 九鼎灿灿黄金砂. (Bai 1988a, p. 147a)[25]

In his only outer alchemical scripture, Bai Yuchan mainly inherited the views of outer alchemy on the tripod. Nevertheless, he and the Southern Lineage of the Golden Elixir gave the tripod, especially the nine tripods, significant status in inner alchemy. However, the further elaborations of the nine tripods made by the Southern Lineage of the Golden Elixir are drawn from a scripture of third-generation disciple, Longmeizi 龍眉子.

## 5. The Return of the Nine Tripods in *Jinye huandan yinzheng tu*

Longmeizi's Daoism came from Bai Yuchan personally, but he also received the teachings of Ruoyizi 若一子, a disciple of Weng Baoguang 翁葆光 (1174–1189), who, in turn, belonged to another lineage of Zhang Boduan. Longmeizi is a fifth-generation disciple of this lineage. His students include Wang Jingxuan 王景玄 and Lin Jing 林靜, whose two aliases are Yuanyangzi 元陽子 and Yaotai xuanshi 瑤臺玄史, respectively (Gai 2013, pp. 535–38).

The *Jinye Huandan yinzheng tu* 金液還丹印證圖 (DZ151) was taught by Bai Yuchan and recorded by Longmeizi. To my knowledge, there are five main editions still extant (Table 1). Overall, the fullest and best version is the one in the *Daoist Canon*, but the other versions have their own unique value. Gai Jianmin has discussed the relationship between the *Daoist Canon* version and the Hanchanzi 涵蟬子 annotated version (Gai 2013, pp. 327–29). According to the diagram and the order of *faxiang* 法象, the version annotated by Lu Xixing 陸西星 (1520–1606)[26] was adapted from the Hanchanzi annotated version. The colored scroll version mainly originated from the *Daoist Canon*, but it may have some connections with the Yiyuzi 頤愚子 poetic version. Though it contains a few obvious mistakes, the Yiyuzi poetic version is a very special version and is earlier than all others, possibly with a source that is different from the *Daoist Canon* version. For instance, the preface of three versions uses two typical Buddhist words—bodhi (*puti* 菩提) and nirvana (*niepan* 涅槃)[27]—to describe the realm of being immortal. This stands in contrast to the *Daoist Canon* version, which is the only source that replaces these terms with *ziran* 自然 (nature) and *taixu* 太虛 (the great void), respectively. When comparing the diagrams of all of the versions, their relationship becomes much clearer. Using the Diagram of the Tripods (*Dingqi tu* 鼎器圖,

Figures 8–11) as an example, the Yiyuzi and colored versions have sawtooth patterns on the sleeves of the female's clothes, and the two tripods have the *Bagua* 八卦 marks of *kan* 坎 and *li* 離 individually. These two marks highlight the association of mutual promotion and mutual restraint with two other marks standing for male and female, *qian* 乾 and *kun* 坤. This relation is ambiguous in the *Daoist Canon* and Hanchanzi versions.

**Table 1.** The Existing Versions of *Jinye huandan yinzheng tu*.

| Version | Date | Title and Author Marks | Source | Prefaces and Postscripts |
|---|---|---|---|---|
| The *Daoist Canon* version | The latest date in the preface is 1249 | 1金液還丹印證圖並序<br>2龍眉子撰 | *Daozang*, vol. 3 (Beijing: Wenwu Chubanshe; Shanghai: Shanghai Shudian; Tianjin: Tianjin Guji Chubanshe), 102c–110a. | 1金液還丹印證圖並序：宋嘉定戊寅仲冬元日龍眉子敘<br>2端平甲午武寧王景玄啓道書<br>3後識[28]：是歲季冬三日用識源流於末<br>4還丹印證圖後敘：己酉歲[29]金精滿鼎日瑤臺玄史元陽子吳興林靜熏潔拜書于太微玄蓋洞天時年三十有五也 |
| | | | *Zhonghua daozang* 中華道藏, vol. 19, no. 092, (Beijing: Huaxia chubanshe, 2004), 714a–723b. | |
| Hanchanzi annotated version | May 1471 (Wang 2012, pp. 72–73) | 1金液還丹印證圖發微<br>2龍眉子圖述<br>3涵蟬子[30]發微 | *Jindan zhengli daquan zhuzhen xuan'ao jicheng* 金丹正理大全諸真玄奧集成, vol. 5, a photocopy of 1538 by *Zhonghua zaizao shanben* 中華再造善本 (Beijing: Guojia tushuguan chubanshe, 2014). | 1金液還丹印證圖序：宋嘉定戊寅仲冬元日龍眉子序<br>2端平甲午武寧王景玄啓道書(啓道號金蟾子)<br>3圖十二幅（系詩十二首），附煉丹行、指迷箴<br>4指迷箴自然明白，茲不贅述，學者味之<br>5後識：是歲季冬三日用識源流於末<br>6後序：己酉歲金精滿鼎日瑤臺玄史元陽子吳興林靜熏潔拜書于太微玄蓋洞天時年三十有五也 |
| | | | *Daoshu quanji jindan zhengli daquan zhuzhen xuan'ao jicheng* 道書全集[31]金丹正理大全諸真玄奧集成, vol. 5. (published in 1519 and repaired by Songxiu Tang 嵩秀堂 in 1682, collected in UC Berkeley) | 1金液還丹印證圖序：宋嘉定戊寅仲冬元日龍眉子序<br>2端平甲午武寧王景玄啓道書<br>3指迷箴自然明白，茲不贅述，學者味之<br>4後識：是歲季冬三日用識源流於末<br>5後序：己酉歲金精滿鼎日瑤臺玄史元陽子吳興林靜熏潔拜書于太微玄蓋洞天時年三十有五也 |
| | | | *Daoshu quanji jindan zhengli daquan zhuzhen xuan'ao jicheng* 道書全集·金丹正理大全諸真玄奧集成 5 (Beijing: Zhongguo shudian, 1990), 389b–412a. (a photocopy of a 1628–1644 version). | |
| | 1796–1820 | 1金液還丹印證圖詩<br>2白玉蟾真人授龍眉子述 涵蟬子註 | *Daozang jiyao* 道藏輯要, no. 6, vol. 11, (Chengdu: Bashu shushe, 1995), 497a–506c. | 1宋嘉定戊寅仲冬元日龍眉子題 |
| Yiyuzi poetic version | A preface written in 1461 | 1金液還丹印證圖次序<br>2龍眉子撰<br>3頤愚子次韻 | Collected in the National Library of China, call number 14257 | 1金液還丹印證圖序：大宋嘉定仲冬九日戊寅龍眉子敘<br>2（金液還丹印證）圖次韻序：時大明天順辛巳暮春望後日 頤愚子述<br>3原道歌 頤愚子<br>4警傍門歌 頤愚子<br>5金液印證還丹圖後序（文末頁佚失，非林靜後序）<br>6素：天順壬午端陽日守素道人[32]書 |
| Lu Xixing annotated version | The annotator lived in 1520–1606 | 1 龍眉子金丹印證詩<br>2 淮南參學弟子陸西星謹測 | *Zangwai daoshu* 藏外道書, vol. 5 (Chengdu: Bashu shushe, 1992), 350a–356b. | 1金液還丹印證圖勘誤 |
| Colored scroll version | Qing dynasty[33] | | Collected in the Baiyun Daoist Monastery (白雲觀) of Beijing | 1宋嘉定戊寅仲冬元日龍眉子敘<br>2端平甲午武寧王景玄啓道書/宋寧宗朝與陳泥丸同時人<br>3金液還丹印證圖，印證圖外法象元章<br>4後識：季冬三日用識源流于末，金液還丹印證圖 |

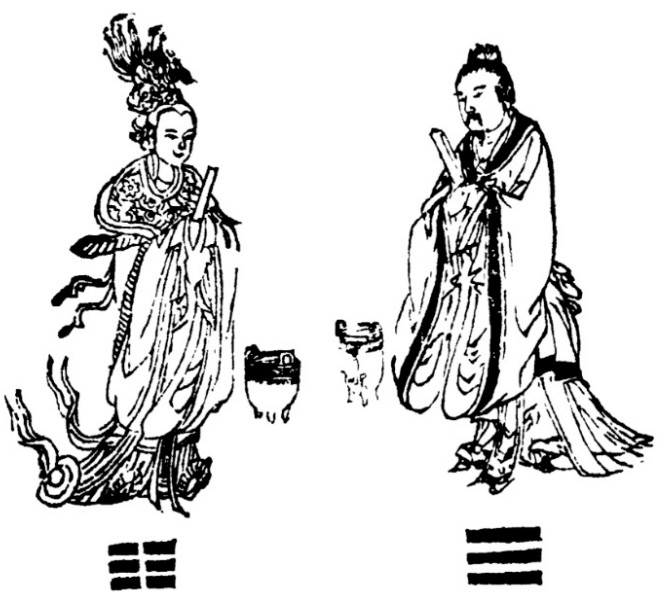

**Figure 8.** The diagram of the tripods found in the *Daoist Canon* version.

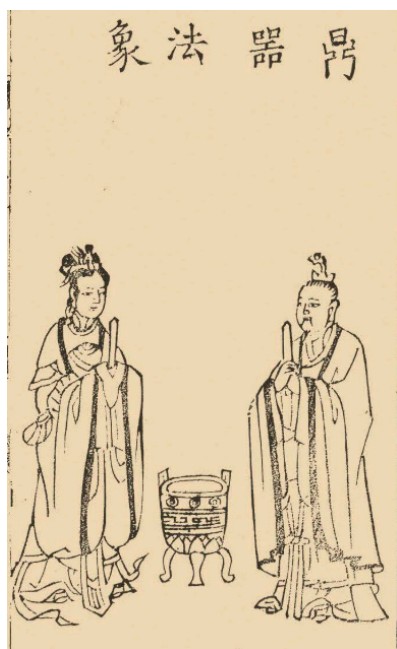

**Figure 9.** The diagram of the tripods found in the Hanchanzi annotated version.

As an important scripture of Daoist inner alchemy, numerous scholars have focused on *Jinye huandan yinzheng tu* (Wang 2007, p. 36; Wang 2012, pp. 61–82; Shi 2015, p. 152). Shawn Eichman indicates that the diagrams represent the process of inner cultivation, noting the differences of the *faxiang* between inner and outer alchemy (Little and Eichman 2000, pp. 344–47). Xu Yilan provides captions for every diagram, applied the three methods of Chen Tuan 陳摶 to read the *Jinye huandan yinzheng tu*, and identified its huge impact on later illustrations of inner alchemy and Visualization and Meditation (*cunsi* 存思) (Xu 2009, pp. 182–217). However, Longmeizi mentions where he learned Daoism in the prologue and epilogue; so, I posit that the emphasis in studying *Jinye huandan yinzheng tu* should fall on how the diagrams express the inner alchemical theory of the Southern Lineage of the Golden Elixir.

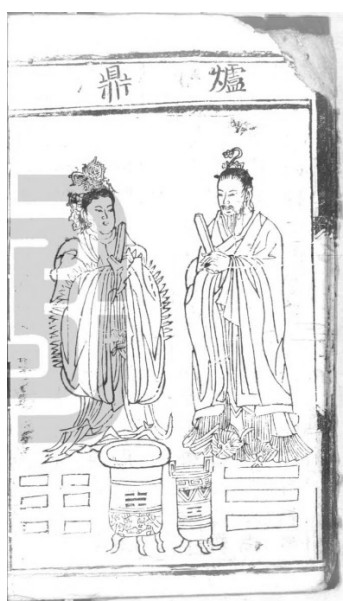

**Figure 10.** The diagram of the tripods found in Yiyuzi poetic version.

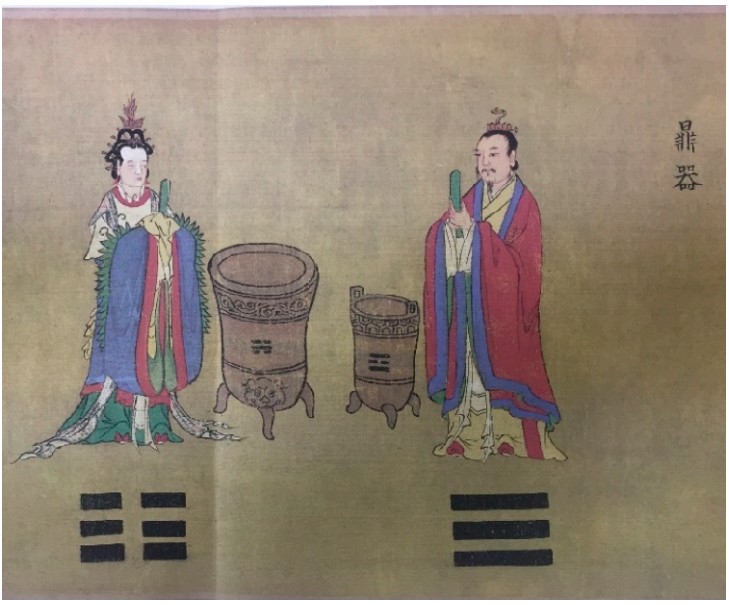

**Figure 11.** The diagram of the tripods found in the colored scroll version[34].

For example, the Southern Lineage of the Golden Elixir thought that the refinement of the elixir is like refining the Essence of Nature (*qi* 炁) by the human body, an essence that existed before the creation of the universe. As an abstract concept, people cannot see the Essence of Nature; they do not even know how to describe it. So, Daoism created a unique Chinese character "炁" to denote this superior essence. The first image of *Jinye huandan yinzheng tu*, *yuanben* 原本 (origin, Figure 12), uses a circle to denote the Essence of Nature in the diagram, rather than the metaphors of outer alchemy—lead and mercury. Before this, few Chinese paintings could accurately portray transcendent concepts. The landscape paintings of Chinese literati use *liubai* 留白, a special technique that deliberately leaves a blank for infinite imagination (Zong 1981, pp. 23–30, 69–88). Calligraphic skill in painting is also used to express the mind of artists (Fong 1984, pp. 74–129). Thus, drawing the transcendent concept directly is an innovation of *Jinye huandan yinzheng tu*. The same method also applied in the last diagram, *huanyuan* 還元 (return to the origin), which constitutes a loop structure of cultivating inner alchemy with *yuanben*.

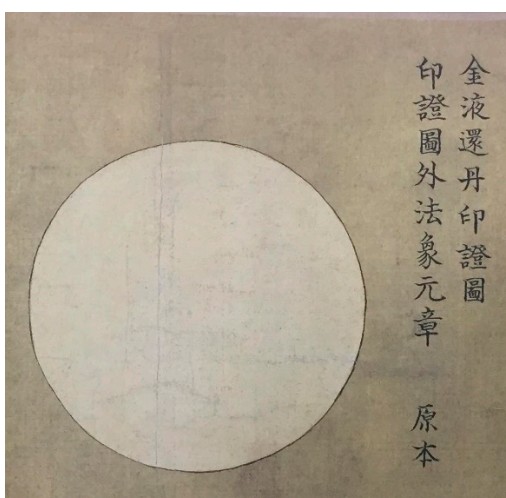

**Figure 12.** Diagram of yuanben in *Jinye huandan yinzheng tu*, colored scroll version.

Other specific scenes are easier to draw than transcendental concepts, but these have no artistic charm and the metaphorical depiction is clumsy. For example, using heaven and the ocean to stand for the difference of *yinyang* 陰陽 and *qiankun* 乾坤; drawing gods, officials, rich people, farmers, animals, and devils in a circle to stand for the Transmigration of the Six Grades (六道輪迴) in *jingwu* 警悟 (warning and understanding); a man and a woman standing with two tripods to reveal that they are equal in *dingqi* (鼎器, tripod); a boy and a girl in two circles with different colors to show the match of genders and the two elements, lead and mercury; and the dragon and tiger taking the elixir, but the tiger possessing a dragon-snake body for holding a balance in the frame.

One of the most common objects in this scroll is the tripod, containing *dingqi*, *caiqu* 採取 (getting), *zhidu* 制度 (regulation),[35] *fuzuo* 輔佐 (assistance),[36] *jiuding* 九鼎, *jinhuo* 進火 (burning), *choutian* 抽添 (distract and add), *muyu* 沐浴 (bath), and *jinye* 金液 (golden liquid), accounting for nine of twenty diagrams. *Jinye huandan yinzheng tu* chooses the traditional tripod instead of the crucible of outer alchemy, just as Bai Yuchan painted in *Jinhua chongbi danjing yaojue*. These tripods are presented using the body-tripod metaphor, but the structure and decorations of traditional tripods are diagramed, such as double ears, three legs, beast faces (*taotie* 饕餮), cicadas, and the decorative lines of cloud and thunder. The note of *dingqi* describes the size and shape of a tripod,[37] but this is a quotation from the *Dingqi ge* of *Zhouyi cantong qi*. Since the Song dynasty, inner alchemy gradually replaced outer alchemy in the utilization of *Zhouyi cantong qi* (Meng 1993, pp. 115–17). As such, this quotation is based on an inner alchemical understanding too.

Among those diagrams containing tripod, the most conspicuous one is *jiuding* (Figure 13). It may be the first time that the nine tripods had been drawn in ancient China. In his preface, Longmeizi classified the *jiuding* as one of nine chapters of "*nei faxiang yangdan* 内法相養丹".[38] The diagram depicts a huge loop composed by nine tripods and five circles with clouds. The poem of Longmeizi describes a certain encounter between Yu and Huangdi:

Evils were terrified since the Emperor Yu molded the tripods; Daoism fulfilled after the Emperor Xuanhuang (Huangdi) cast tripods. 帝禹範來奸始怖，軒黃鑄就道方成.[39]

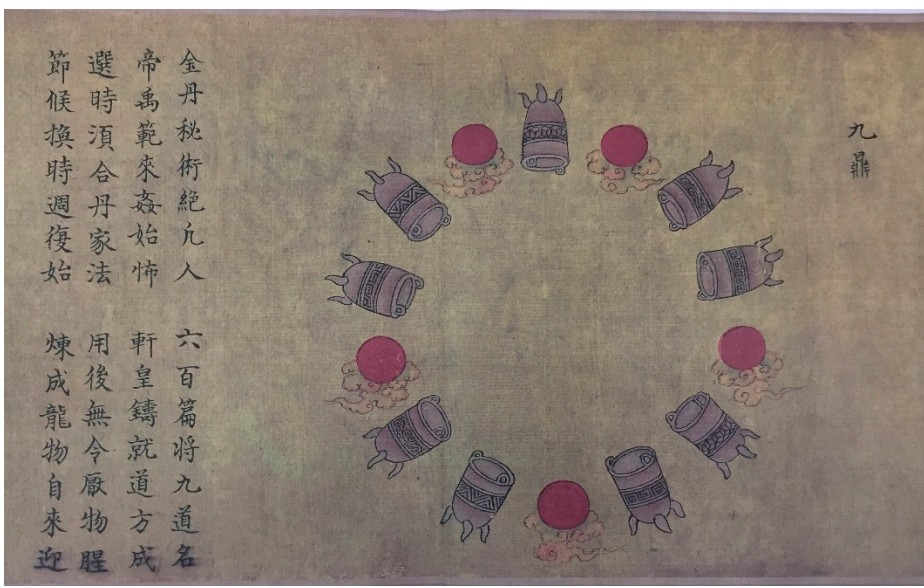

**Figure 13.** Diagram of the nine tripods, *Jinye huandan yinzheng tu,* colored scroll version, collected by Baiyun Daoist Monastery of Beijing.

Yu's contribution to the tripod was finally recognized by Daoism. In addition, the second sentence of Longmeizi's poem refers to a numerical concept related to the "nine" of the nine tripods:

Six hundred chapters fulfil the nine orbits. 六百篇將九道名.

Yiyuzi notes that the tripods have the number nine because the land under the sky (*tianxia* 天下) is split into nine states (*jiuzhou* 九州). But other annotators have different opinions. Hanchanzi explains that refining elixir needs three hundred days and nights according to *Huoji* 火記 (the records of fire). *Huoji* is a scripture that expounds the duration of refining elixir, which was introduced by *Zhouyi cantong qi* in its twenty-nineth chapter. Zhang Boduan also had a poem called *Huoji liubai pian* 火記六百篇. According to the introduction of *Zhouyi cantong qi*, Hanchanzi provided further explanation about the duration of fire and the *bagua*. Tripods and the medicines (*dingqi yaowu* 鼎器藥物) of elixir are made up of four of the *bagua*'s sixty-four elements, and the other sixty elements represent the time for refining elixir. He calculated that the whole process of refining elixir needs 300 days ([Bai 2013a](), p. 156). Lu Xixing explained further that the 300 days are ten months, and every month expends sixty elements.[40] Every day uses two elements in morning and evening, so that is six hundred elements (*liubai gua* 六百卦).

Regarding *jiudao* 九道 (nine orbits), Hanchanzi thought that it is just the nine tripods, because refining the elixir needs nine cyclical transformations.[41] Lu Xixing, for his part, engages in a deductive process: *jiudao* is the orbits of sun and moon, with the sun running in one orbit and the moon in eight orbits. In the 300 days of refining elixir, the sun and moon run a complete cycle of nine orbits. So, the "nine" of the nine orbits indicates the process of refining elixir rather than the number of tripods. The nine refers to nine periods out of the twelve two-hour periods of a day (*shier shichen* 十二時辰), from *zi* time (*zi shi* 子時) to *xu* time (*xu shi* 戌時), excepting the *mao* and *you* times (卯酉) for rest.[42]

On the other hand, the *Daoist Canon* and the colored scroll versions draw five circles with clouds between the nine tripods. Two of the circles clamp a tripod in the middle of the upward side. The other three circles attach tripods in the middle of other three directions. But the versions annotated by Hanchanzi and Yiyuzi paint nine circles alternating the nine tripods. All of the annotators failed to explain explicitly what the circle means. Nevertheless, there are clues that can help us infer this meaning. First, the earliest version paints the five circles in specific places. The seventh sentence of Longmeizi's poem, "restart the loop in the changing time of seasons" (節候換時周複始), may suggests that the loop

structure also pertains to season changing. So, Xu Yilan considers that the five circles are the five elements (*wuxing* 五行) and five stars (Xu 2009, p. 186). However, according to the comment of Lu Xixing about the orbits of sun and moon running, the five circles could represent five moments of the movement of the sun and moon. There is a sentence in *Wuzhen pian*, "between the two half-moon days . . . since the elixir refined, the cultivator has to incubate and nourish it like cooking fresh fish 前弦之后后弦前 . . . . . . 炼成温养似烹煎" (Zhang and Wang 1990, pp. 81–82),[43] referring to the lunar phases of refining elixir. Weng Baoguang explained the explicit relations among lunar phases, *bugua,* and refining elixir (Weng 1988, p. 16). Bai Yuchan's *Gousuo lianhuan jing* also elaborated them (Bai 2013a, p. 168).

There are five time points for every lunar month, the first day (*shuo* 朔), middle day (*wang* 望), last day (*hui* 晦), and the two half-moon days (*xian* 弦). The half-moon days include the first quarter and the last quarter in 8 and 23. These are drawn on the left and right sides of the diagram of nine tripods. Weng Baoguang said that "in the last day and first day of two months, the moon intersects with the sun. They run together, appear together, and disappear together 此时与日相交，在晦朔两日之中，合体而行，同出同没" (Weng 1988, p. 16b). So, the two circles in the upward trajectory should be the last day of the previous month and the first day of the next. Moreover, the scribe of the colored scroll version painted the circle in red, which is probably the color of the moon. The cloud surrounding the moon would make sense then. In short, the circles represent the moon at the five time points of a month rather than the sun.

Thereafter, the innovations in Longmeizi's tripod set a model for the inner alchemy of later times. For example, *Xingming guizhi* 性命圭旨, which is an inner alchemical manual published at the end of Ming dynasty (Wu 1615), depicts the nine tripods in the "diagram of refining the heart through the nine tripods" (*Jiuding lianxin tu* 九鼎煉心圖, Figure 14) (Wu 1615, p. 1.41). This diagram illustrates nine different shapes of tripods, which correspond to the nine hearts with different degrees of cultivation. Depicting heart refining is carried out through the lunar phases, suggesting a consistency with the diagram of nine tripods in *Jinye huandan yinzheng tu*. Here, the nine tripods, the nine cyclical transformations, and the lunar phases are integrated in one diagram.

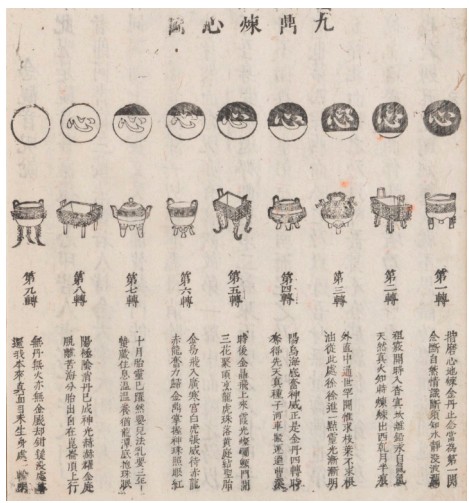

**Figure 14.** Diagram of refining the heart by the nine tripods (Wu 1615, 1.41).

The second example of the influence of Longmeizi's tripod is the theory of *sanguan* 三關 (three gates). *Sanguan* previously indicated the three gates of river chariot (*heche sanguan* 河車三關)—three organs or acupoints of human body (Li and Jiang 2010, 3.44, 182; Zeng 2016, pp. 48–50, 66)—or form (*xing* 形), pneuma (*qi* 氣), and spirit (*shen* 神) (Bai 2013b, p. 56). Wu Chongxu 伍沖虛 (1574–?), a Daoist at the end of Ming dynasty, bestowed the *sanguan* to three concepts of time, the hundred days gate (*bairi guan* 百日關), the ten months

gate (*shiyue guan* 十月關), and the nine years gate (*jiunian guan* 九年關) (Wu and Liu 2013, pp. 149, 159–63). According to Wang Mu, the "ten months" and "nine years" indicate, respectively, the period of pregnancy and the time that Bodhidharma, the first patriarch of Ch'an is reputed to have gazed at the wall of his cave (Wang 2008, pp. 157–58, 67).[44] But he might have taken the words too literally. Wu Chongxu was just developing the theory of tripod found in *Jinye huandan yinzheng tu*.[45]

 *Qiangong jiageng zhibao jicheng* is an outer alchemical collection compiled in 1316 or 1366, which depicts two traditional tripod shaped refiners. Its editor, Zhao Naian 赵耐 庵, who lived at the end of Yuan dynasty and the beginning of Ming dynasty, called the inside part of the first refiner the "iron rack with three legs" (*tie sanjiao jiazi* 铁三角架子, Figure 15) and the second refiner the tripod (Figure 16). They both look like the traditional tripod, but the first one is the interior structure of a double structured refiner,[46] while the second one has two parts, a top tripod (*shangding* 上鼎) and a bottom tripod (*xiading* 下鼎). In that sense, these two tripods still maintained the double structured refiner found in outer alchemy. Aside from that, in the notes, the structure and size of these refiners carry meanings associated with time, such as the twelve double-hours (*shier shichen* 十二時辰), the eight festivals (*bajie* 八節), and the twenty-four solar terms (*ershisi qi* 二十四氣). In addition, Zhao Naian called water, fire, and medicine as the *sancai*, as Ye Wenshu did (Zhao 1988, pp. 260–61). All of these instances show that this outer alchemical scripture was influenced by inner alchemy. Zhao claimed the scripture was inherited from Zhang Fuhu 張富壺 and Yang Jiuding 楊九鼎 of Sichuan, but they and Zhao are hard to confirm historically.[47] In the Jiading period (1207–1224) of the Southern Song dynasty, Sichuan had a financial official named Yang Jiuding (Lu 1991, p. 334), but there is no record to show that he was an alchemical master. The name of "*Fuhu*" looks like a Daoist word and ostensibly refers to the *penghu*, but, to my knowledge, this is the only case it appears in Daoist or secular texts. Wang Jiakui claims that Zhao Naian falsified much of the content in his book (Wang 2000). Thus, the two masters may have been invented by Zhao Naian too. The name of Yang Jiuding is another clue that this outer alchemical scripture could have been influenced by the inner alchemy of the Southern School of the Golden Elixir.

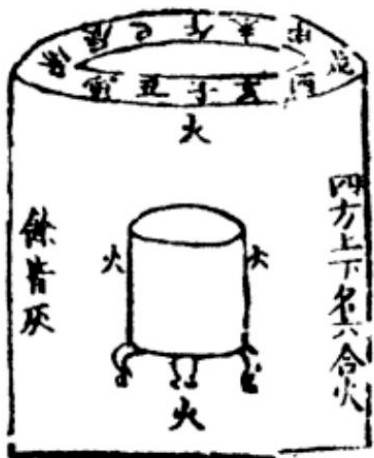

**Figure 15.** The cabinet of the fountain and the iron rack with three legs (Zhao 1988, p. 249c).

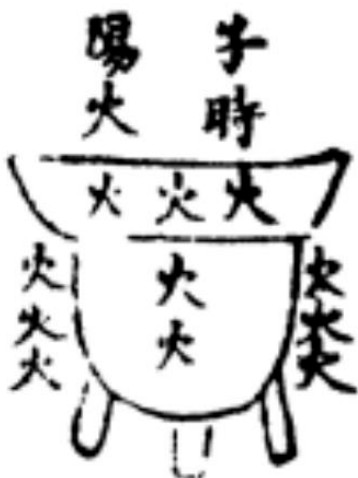

**Figure 16.** The method of using the tripod (*yongding fa* 用鼎法) (Zhao 1988, p. 260c).

## 6. Conclusions

Overall, the tripod and the nine tripods experienced two variations in the two stages of Daoist alchemy, from a mysterious object to a refining facility and back to a mysterious and holy object. In the beginning, tripod and the nine tripods were part of tales and rituals in ancient China. The name and tale of tripod were introduced in the early scriptures of outer alchemy. But outer alchemy developed the double structured crucible for refining elixir more efficiently, giving up the traditional shape of tripod. Inner alchemy created a body-tripod metaphor based on outer alchemy's processes for refining elixir using this implement. Recognizing that the significance of tripod and the nine tripods had been neglected, Bai Yuchan and his disciples reintroduced this concept during the Southern Song dynasty, establishing the unique relations between the lunar phases, the tripod, and the process of refining elixir that would influence later Daoist alchemy. In this article, I traced how the two traditions—outer and inner alchemy—neglected and rebuilt the tripod and the nine tripods.

Inner alchemy's reconstruction of the significance of the tripod and the nine tripods not only borrowed the knowledge from outer alchemy, but also drew from non-Daoist resources. For example, Bai Yuchan cites an astronomical book to annotate a scripture under the name of Laozi 老子 (Bai 2013a, p. 193). The Southern Lineage of the Golden Elixir school's understanding of the nine tripods also exhibits the knowledge of astronomy. A second possible external source is antiquarianism, which is a fashion or scholarship of collecting and studying the ancient bronze vessels that rose during the Song dynasty (Chen 2016). The literati of the Song dynasty interpreted the rituals and words of ancient times through the study of bronze vessels. The diagrams of tripods of inner alchemy are often painted with the archaic decorations, may displaying the influence from this antiquarianism.

**Funding:** This research received no external funding.

**Institutional Review Board Statement:** Not applicable.

**Informed Consent Statement:** Not applicable.

**Data Availability Statement:** Not applicable.

**Conflicts of Interest:** The author declares no conflict of interest.

## Notes

[1]  "We have heard that in ancient times the Great Emperor Fu Xi made a single sacred cauldron. The number one symbolizes the unification of heaven and earth, showing that all things of creation were brought together. The Yellow Emperor made three

precious cauldrons, symbolizing heaven, earth, and man, while Emperor Yu collected metal from the nine ancient provinces and cast nine cauldrons" (Sima 1993, p. 34).

[2] *Wudi benji* 五帝本紀 reflects an ideal model of the sovereign in Sima Qian's mind, but *fangshi* depicted Huangdi as a god. See (Marsili 2003; Le Blanc 1985).

[3] In the Tang dynasty, this scripture was renamed *Huangdi jiuding shendan jing* 黃帝九鼎神丹經 as the first chapter in an expanded book, entitled *Huangdi jiuding shendan jingjue* 黃帝九鼎神丹經訣 (DZ885). See (*Huangdi Jiuding Shendan Jingjue Jiaoshi* 2015, pp. 3–9; Pregadio 2006, pp. 241–54). Moreover, Gongsun Qin 公孫卿 claimed Shengong 申公 had received a *Dingshu* 鼎書 (tripod book) from Huangdi, but comparing the few words recorded by *Shiji* and the chapter one of *Huangdi jiuding shendan jingjue*, they turn out to be quite different.

[4] For two recent literature reviews, see (Han 2009, pp. 373–86; Pregadio 2019).

[5] Bai Liang had briefly introduced the cultivating method of Li Daochun 李道純's theory of tripod, but he did not discuss about the symbolism of tripod (Bai 2008).

[6] Ge Hong pointed out that the making of the Great Clarity Elixir is harder than making the nine tripods. Wang Ming cited two books as evidence that the nine tripods correspond to the nine elixirs originally. See (Ge 1996, pp. 76–77, 98).

[7] They are *Taishang dongxuan sandong kaitian fenglei yubu zhimo shenzhou jing* 太上洞玄三洞開天風雷禹步制魔神咒經 (DZ385) and *Longruiguan yuxue yangming dongtian tujing* 龍瑞觀禹穴陽明洞天圖經 (DZ604).

[8] Fabrizio Pregadio noticed this problem (Pregadio 2006, p. 281).

[9] This tripod retains the structure of three legs of the traditional bronze tripod but lacks the container part. See (Feng 2010, pp. 132–33; *Huangdi Jiuding Shendan Jingjue Jiaoshi* 2015, pp. 95, 97–98).

[10] Needham listed some cases of the use of the alchemical crucible, but their time and function are usually mismatched with outer alchemy (Needham et al. 1980, pp. 13–15, 20–21, 23–32, etc.).

[11] The volume one and three of *Qiangong jiageng zhibao jicheng* 鉛汞甲庚至寶集成 (DZ919) drew two diagrams of tripod with three legs, but this scripture was edited in 1316 or 1366. See (Wang 2000). This paper will discuss it in Section 5.

[12] Zhang Guangbao thinks that there is solid evidence that inner alchemy began to thrive from the time of Ye Fashan 葉法善, a Daoist who lived in Tang Xuanzong period (712–756) (Zhang 2001, pp. 50–55).

[13] On the authenticity of this scripture, see (Zhongli and Lü 2015, pp. 17–22).

[14] "Their shapes resemble that of a pot" (Foster 1977, p. 395).

[15] Xiao Hanming explained that *peng* is the crucible and *hu* is the tripod, but he has mistaken the term for another metaphor and lacks evidence. See (Xiao and Guo 2001, p. 293).

[16] Needham acknowledges that the Suspending Fetus Tripod could be both used in inner and outer alchemy, but he thinks that all the diagrams belong to inner alchemy (Needham and Lu 1983, pp. 97–107).

[17] Gai Jianmin is certain that the scripture was created by Bai Yuchan. Meanwhile, Zhan Shichuang hypothesized that two poems in the scripture may have been written by Peng Si 彭耜, a disciple of Bai who lived in the Southern Song dynasty. However, Yokote Yutaka judged that it is a fake book because Bai Yuchan denigrated outer alchemy in a dialogue with Peng Si. Still, because Bai Yuchan puts down outer alchemy in only one sentence, it is hard to deny the authenticity of *Jinhua chongbi danjing mizhi*. Along the same lines, Farzeen Hussein argued that the signed author Lan Yuanbai 蘭元白 of the second volume is another Daoist (Lan Yuandao 蘭/元道 or Lan Fang 藍方), rather than Bai Yuchan, but she did not provide more details. See (Zhan 2002; Gai 2013, pp. 136–38; Yokote 1996; Hussein 2005a).

[18] There is a possibility of that chapter 2 in *Jinhua chongbi danjing mizhi* represents a different viewpoint because it has a different author. See (Hussein 2005a).

[19] Mao Liya agrees that *Yinfu sui* was written by Chen Nan too, though she did not carry out textual criticism (Mao 2004).

[20] Gai Jianmin proved these sketches belong to *Jindan jiejing* 金丹捷徑, a part of *Haiqiong chuandao ji* (Gai 2013, pp. 199–202).

[21] Gai Jianmin thought the "nine orifices" (*jiuqiao* 九竅) is a mistaken writing of "*kongqiao* 孔竅", but he does not provide evidence (Bai 2013a, p. 57). I think that it is not wrong to assume that nine corresponds to the "nine" of nine tripods.

[22] The *Haiqiong chuandao ji* version of the *Daoist Canon* does not have the two tickles but adds a word "*zhi* 之" in the title of diagram. Meanwhile, the punctuated version of Gai Jianmin paints two diagonals instead of points (Bai 1988a, p. 6b). I am not certain about his source.

[23] Bai Yuchan wrote three articles to record his visiting in the Yulong Wanshou Daoist Monastery. The other two articles are *Longsha xianhui ge ji* 龍沙仙會閣記 and *Yulong wanshou gong yunhui tang ji* 玉隆萬壽宮雲會堂記. See (Bai 2013a, pp. 236–37).

[24] Excepting a very late inscription of the Qing dynasty, only Bai Yuchan recorded Luo Ruoxu's name in the article. His name is Ruoxu and the nickname is Shian 適庵. Bai called him an alchemist (*lianshi* 煉師) (Xu 2014, pp. 719–21; Bai 2013a, pp. 237, 40).

[25] Gai Jianmin considered the poem to be written by a disciple of Bai Yuchan (Bai 2013b, pp. 208–16).

[26] On the relationship between Lu Xixing and this book, see (Mozias 2020, pp. 44–45, 176–79).

[27] Daoism generally use *niwan* (泥丸) to translate nirvana, but the preface of Longmeizi still use *niepan*, the Chinese Buddhist translation for nirvana. See (Maspero 1981, p. 457).

[28] The afterword lacks the signature and time, but it was written in the first person and the latest time in the article is very close to the writing time of the preface. Gai Jianmin indicated that the author of this afterword might be the author of the book, Longmeizi (Gai 2013, p. 328).

[29] *Yiyou* has many options; Gai Jianmin argued that the likely time of composition of this afterword is 1249.

[30] Hanchanzi was a Daoist who lived in Zixia Mountain (紫霞山) of Zunyi at the end of the Yuan dynasty and the beginning of the Ming dynasty. See (Zhonghua Zaizao Shanben Gongcheng Bianzuan Chuban Weiyuanhui 2017, pp. 326–27).

[31] On the relation between *Jindan zhengli dacheng* and *Daoshu quanji*, see (Kim 1995).

[32] Shousu Daoren is Yao Fu 姚福 (1427–?), alias Shichang 世昌, as a commander of the one thousand soldiers of the palace guard army (*yulin jun qianhu* 羽林衛千戶). He wrote *Qingxi xiabi* 青溪暇筆, collecting stories from the Hongwu to the Chenghua periods (1368–1487). Perhaps Yiyuzi is Yao Fu. See (Huang 2001, p. 334).

[33] Eichman argues that the scroll was painted in the Qing dynasty, while Wang Yie thinks it was painted in the Ming dynasty, but they do not provide evidence (Little and Eichman 2000, p. 131; Wang 2011, p. 179). There is no sign or seal in the scroll showing the time in which it was drawn. The copy time and copier deserve more discussion, and nobody has debated the source of the color until now.

[34] Thanks to Mrs. Wang Yie for allowing me to photograph and use this scroll. For the full scroll, see (Little and Eichman 2000, pp. 346–47).

[35] Only in the Hanchanzi annotated version.

[36] Only in the Hanchanzi annotated version.

[37] "Five-three in circumference and one in diameter, four-eight in mouth and the navel apply in abdomen 圓繞五三圍徑一，脣周四八腹臍敷". It changed the original text, "3 and 5 around, an inch and one part, 4 and 8 the mouth, the lips a pair of inches 圓三五，徑一分；口四八，兩寸脣". See (Pregadio 2011, p. 120).

[38] In the colored scroll version, "相" was changed to "象".

[39] "*Fang* 方" in the Hanchanzi and Lu Xixing annotated versions is "*ming* 名".

[40] "The six hundred chapters of the *Record of Fire* are the number of ten months' trigrams … every month has sixty trigrams, so the ten months have six hundred trigams, which is the final number of trigrams and fire 火記六百乃十月之卦數 …… 每月計卦六十，十月六百，而卦火之數終矣" (Longmeizi and Lu 1992, p. 353b).

[41] "The nine orbits are the nine tripods … the nine tripods mean the nine cyclical transformations. Nine is a positive number, which indicates that the largest number is nine 九道名即九鼎也 …… 九鼎，乃九轉之義，陽數，極於九而言" (Bai 2013a, p. 156).

[42] "*Jiudao* is the orbit of sun and moon running. The astronomer said: 'sun runs in the middle orbit and moon runs in eight orbits.' Alchemy is like the sun and moon, so the fire of ten months like the sun and moon running in the nine orbits … the ancient immortals refined and cultivated the nine tripods, so the nine tripods are the nine orbits. Why are the efforts of ten months called nine tripods? Because starting from Zi time, the warehouse of fire will return in Xu time. That is, except for bathing in Mao and You time. So, the nine cyclical transformations are named for the nine tripods 九道乃日月運行之跡道，曆象家云：日行中道，月行八道。丹法象日象月，故十月之火，如日月運行於九道之中 …… 然而上古仙人服煉九鼎，九鼎之說即九道也。丹有十月之功而曰九鼎，何也？曰，自子至戌，而火庫歸矣，又除卯酉沐浴而言之，凡有九轉故曰九鼎" (Longmeizi and Lu 1992, p. 353b).

[43] Tenney Davis and Thomas Cleary translated it into English, but their translations of some words are inaccurate (Davis and Chao 1939; Chang and Liu 1987).

[44] Wu Chongxu answered that the facing the wall nine years is an elementary process to becoming immortal, but he did not refer to the name of Bodhidharma (Wu and Liu 2013, pp. 118–19, 163).

[45] Longmeizi said that "one must incubate and nourish oneself after taking elixir, so it has the nine tripods 服畢務溫養，故有九鼎焉" in the preface, but it is hard to confirm that there is a direct relation between the nine tripods and the nine years gate (Bai 2013a, p. 125).

[46] Chen Guofu confirmed that the cabinet of the fountain (*yongquan gui* 湧泉櫃) was created in the Tang dynasty (Chen 1983, p. 340).

[47] Zhao Naian, alias Zhiyizi 知一子. The relationship between Zhao and Ruoyizi 若一子 (one of the masters of Longmeizi) is hard to be sure of.

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
