# Peer review of "The Tripods in Daoist Alchemy: Uncovering a Material Source of Immortality"

_religions, doi:10.3390/rel13090867_

Round 1

Reviewer 1 Report

This is an excellent article that I recommend for publishing.

Here are some suggestions for enhancement of the article.

Eugene Wang authored a very important article about the function of ding as an agent to nurture breath, perhaps the author wishes to mention it. See Eugene Wang Afterlife Entertainment? The Cauldron and Bare-torso Figures at the First Emperor's Tomb. This article shall be mentioned and quoted.

The author listed predominantly Chinese literature; perhaps he/she might like to mention a few more English literature, such as Susan Huang’s Daoist Visual Culture and Stephen Little’s Taoism and the arts of China.

Author Response

Dear reviewer,

Thank you very much for liking my article and reminding me of the three important references.

The paper of Prof. Eugene Wang is included in a conference proceeding, which is indeed difficult to notice. This is an imaginative treatise that uses the tripod as a clue to speculate on the time and cosmic system in the First Emperor's tomb. Prof. Wang even connects the tripod of the Qin dynasty to an inner alchemical book of the Ming dynasty. There is a huge gap in time. I cannot accept it for now and I think the great paper would not affect the main judgments of my article. I add this paper in the literature review part, please see Line 59–62 of the revised version.

Susan Huang’s book is one of the most important monographs in the field of Daoist art. The book always stays on my desk and actually I cited it in the earliest version of this article. Prof. Huang studies the Jinye huandan yinzheng tu in her book, but she does not pay more attention to the tripod of this painting. Therefore, I removed it from the reference for the principle of precise exposition.

About the book edited by Stephen Little and Shawn Eichman, this is an important reference and I have listed it. Please see Line 255.

Sincerely,

Reviewer 2 Report

Hi

The paper needs some English editing. You could start with a grammar check. it would be best if a native English writer correct the grammar, word choice, and use of articles (a/the).

I prefer the expression "external alchemy" instead of "outer alchemy". The contrast of internal vs external is a smoother expression.

Author Response

Dear reviewer,

Thank you for the suggestions on language and I will work on it.

About the translating choice of external/internal and outer/inner alchemy, actually I chose the external/internal alchemy in the earliest version, and I found the two choices are both existing in the English papers about Daoist alchemy. However, via careful thinking and referring to the translations in the Encyclopedia of Taoism edited by Fabrizio Pregadio (2008, p. 762, 1002), who is the authoritative scholar in the English studies of Daoist alchemy, I decide to use outer/inner alchemy translating waidan/neidan of Chinese. Please understand my choice.

Sincerely,